# Parental Lifestyle Changes and Correlations with Children’s Dietary Changes during the First COVID-19 Lockdown in Greece: The COV-EAT Study

**DOI:** 10.3390/children9121963

**Published:** 2022-12-14

**Authors:** Georgios Saltaouras, Maria Perperidi, Christos Georgiou, Odysseas Androutsos

**Affiliations:** 1Laboratory of Clinical Nutrition and Dietetics, Department of Nutrition and Dietetics, School of Physical Education, Sport Science and Dietetics, University of Thessaly, 42132 Trikala, Greece; 2Department of Nutrition and Dietetics, School of Physical Education, Sport Science and Dietetics, University of Thessaly, 42132 Trikala, Greece

**Keywords:** COVID-19, lifestyle, parents, determinants, diet, physical activity, COV-EAT

## Abstract

The aim of this study was to investigate changes in the eating behaviour of parents during the first lockdown implemented in Greece due to COVID-19 and to explore possible associations with corresponding changes in the eating behaviour of their children. A quantitative cross-sectional study was performed using an online questionnaire. The study sample consisted of 397 parents with children aged 2–18 years, who were recruited from 63 municipalities in Greece. It was observed that the percentage of parents and children reporting consumption of breakfast during the lockdown period increased by 10.6% and 5%, respectively. Also, 75% of the parents increased their snack consumption and 61% their sweets consumption. Parents increased home-cooking during lockdown (6.4 times/week), compared to 5.6 times/week before (*p* < 0.001), which was associated with decreased consumption of fast foods for both parents and children (*p* < 0.001 for all comparisons) and also correlated with increased consumption of fruit and vegetables for children (*p* < 0.05). More than half parents tried to lose weight during lockdown (58.4%). In conclusion, both favourable (home-cooking) and unfavourable (increased snacking) lifestyle changes during the first COVID-19 lockdown in Greece were reported for parents.

## 1. Introduction

The coronavirus disease 2019 (COVID-19), induced by severe acute respiratory syndrome coronavirus 2 (SARSCoV-2), is a respiratory infectious disease that broke out in December 2019 and rapidly spread around the world, leading the World Health Organisation declaring a pandemic state [1]. Most governments worldwide imposed isolation measures as a way to control the spread of the disease, including the closure of schools and restaurants, work-from-home policies, travel bans and lockdowns—partial or complete, especially during the beginning of the pandemic. The impose of social isolation measures due to the COVID-19 pandemic in 2020 globally has had unprecedented consequences in people’s physical and mental wellbeing [2]. In Greece, first complete lockdown, along with other containment measures, was imposed in March 2020.

It is known that containment measures have affected people’s lifestyle, including alcohol consumption [2], smoking [3], physical activity [4,5], sedentary time [6], sleeping quality [5] and dietary habits [7,8,9]. Regarding diet, several published reviews to date, have summarised the effects of containment measures (lockdowns) in people’s diet, including changes in food intake, eating behaviours, diet quality [8], snacking [8,10] and adherence to healthy dietary patterns, such as the Mediterranean Diet [9]. Available evidence is, however, conflicting and findings may be driven by demographic and socioeconomic factors, such as gender, employment and ethnicity [11]. A lot of work has also focussed on dietary and physical activity of children during the first lockdown [12,13,14]. An earlier study in children and adolescents in Greece has shown changes in lifestyle behaviours during the first lockdown (March–May 2020), such as increased sedentary time, reduced physical activity, increased consumption of sweets but also increased consumption of fruit, juices, vegetables, dairy and decreased consumption of fast food [12].

The family is an important social context where children adopt eating behaviours and develop attitudes towards food. The role of parents/guardians in the quality of a child’s diet is well documented in the literature, as they play the role of health promoters, role models and educators, influencing their food choices [15,16]. Dietary changes and practices of parents/guardians and their influence on children’s lifestyle behaviours during the pandemic are not known. A recent, small study in South Australia showed that there was no change in total energy intake or body weight in parents during the first lockdown; however, an increase in energy from alcohol was found [17]. It could be assumed that the impose of containment measures would influence the dietary habits and meal preparation practices of parents/guardians which, in turn, could influence the dietary habits of their children. The aim of this study was to investigate changes in the eating behaviour of parents during the first lockdown implemented in Greece due to COVID-19 and to explore possible correlations with corresponding changes in the eating behaviour of their children.

## 2. Method

### 2.1. Study Design and Participants

The COV-EAT study design has been described in detail in a previous publication [12]. Briefly, a cross-sectional study was conducted across 63 municipalities in Greece between March–May 2020 (first lockdown in Greece), in which parents with children between the ages of 2 and 18 years were invited to take part. Eligibility criteria included: living in Greece, being able to complete the study questionnaire in Greek language, having children aged 2–18 years, and providing a consent form. The COV-EAT study was conducted in accordance with the Declaration of Helsinki, and the protocol was approved by the Ethics Committee of the Department of Physical Education and Sport Science in the School of Physical Education, Sport Science, and Dietetics, University of Thessaly, and registered at clinicaltrials.org (NCT04437121). All subjects digitally provided an informed consent for inclusion before participation in the study.

### 2.2. Instruments and Variables

The COV-EAT study incorporated an online survey, which included 70 multiple-choice questions in total, divided in 3 sections:

Section 1: Parents’ socio-demographic and anthropometric data, including area of residence, education, employment, family status, as well as age, height, body weight and weight changes for both parents and children.

Section 2: Parents’ lifestyle and dietary changes before and during first lockdown due to COVID-19, including frequency of cooking at home and fast-food consumption, number of meals and snacks consumed daily and reasons for snacking. Also, questions regarding interest for health eating and weight loss practices were included.

Section 3: Child’s eating habits, such the frequency of breakfast, fast-food, fruits, juices, vegetables, dairy, red meat, poultry, fish, pasta, legumes, sweets, salty snacks, and beverages consumption and vitamin supplementation. Data for only one child per family was included in this study.

Participants were asked to provide answers about the period before and during the first lockdown. Differences between the values of each behaviour were categorised by the researchers as “decrease” if the value of behaviour was higher before lockdown compared to the value during lockdown, or “increase” if the value of behaviour was lower before lockdown compared to the value during lockdown. If the value of behaviour remained stable before and during lockdown, it was categorised as “stable”.

The questionnaire was created with the use of Google forms. The distribution of the survey was done via networks of dietitians and nutritionists in Greece, through social media and personal networks. Data were collected between 30 April and 24 May 2020.

### 2.3. Statistical Analysis

Continuous data are presented as mean ± standard deviation (SD). Categorical variables are presented as absolute (n) and relative (%) frequencies. The variables “preparing meal at home” and “ordering fast food” were transformed from ordinal to continuous. Mean values of frequency of preparation of home meals and consumption of fast foods before and during lockdown were compared using the Student’s test for paired data. For paired pre- vs. post-comparisons of categorical data (consumption of breakfast, lunch, dinner and number of snacks per day), the Wilcoxon matched-pairs signed-rank test was applied. Spearman’s rho was used to explore correlations of changes in frequency of home-cooked meals and children’s consumption of food groups. The level of statistical significance was set to *p* < 0.05 for analyses. All analyses were performed with SPSS V26 software package (IBM, Armonk, NY, USA).

## 3. Results

A total of 397 completed questionnaires were received, which were completed predominantly by females/mothers (89.9%). Table 1 and Table 2 present the anthropometric and socio-demographic characteristics of the parents, respectively. In general, fathers’ age and Body Mass Index were higher than mothers’. More than half respondents (55.7%) reported two children in the household. The majority of respondents reported being married, being employed (both mothers and fathers), having a University qualification (both mothers and fathers) and living in an urban area. More than half fathers (54.4%) and mothers (58.2%) reported a change in their employment status.

Table 3 presents the changes in parents’ practices in relation to food and nutritional supplements. The vast majority of respondents reported eating all three main meals (breakfast, lunch and dinner), both before and during lockdown. However, the percentage of respondents who reported eating breakfast increased significantly during lockdown (83.1% vs. 93.7%, *p* < 0.001), while report of eating lunch and dinner remained similar. A higher percentage of children was reported to consume breakfast during lockdown, compared to before lockdown, as well (89.9% vs. 94.9%, *p* = 0.002) (Figure 1). There was a trend towards increasing use of dietary supplements in children, but did not reach statistical significance, except for the use of vitamin D supplements (Table 3).

Three out of four parents increased snacking during lockdown (very little 18.6%, a little 26.0%, considerably 25.9%, immensely 4.5%) compared to the period before lockdown. Almost a third of parents reported having at least three snacks per day, compared to 8.1% before lockdown. Reasons for increased snacking included boredom (43.1%), stress (24.7%), lack of sleep (16.9%), feeling hungrier (16.9%) and the need to boost the immune system (3.5%). Participants reported eating sweets (chocolate, biscuits etc.; 60.5%), nuts (46.9%), fruit and vegetables (44.3%), grains (crackers, cereals, bread etc.; 41.6%), salty snacks (crisps, popcorn etc.; 35.5%), dairy products (milk, cheese, yoghurt; 25.2%) and processed meat (ham, salami etc.; 14.9%) for snacking.

Participants also reported increasing preparation/consumption of home-cooked meals during lockdown to 6.4 times/week, compared to 5.6 times/week before (*p* < 0.001), predominantly prepared by the mother (84.9% before and 92.9% during lockdown), while decreasing consumption of fast foods (1.1 times/week before Vs 0.6 times/week after, *p* < 0.001). Only five participants (1.2%) reported reducing the frequency of preparing home-cooked meals. The increase in preparation of home-cooked meals was associated with a significant decrease in the consumption of fast foods (*p* < 0.001). Similar to parents, the frequency of consuming fast foods for children decreased significantly during lockdown (0.9 times/week to 0.6 times/week; *p* < 0.001). The decrease in parents’ consumption of fast foods was associated with a decrease in children’s consumption of fast foods (*p* = 0.005).

Table 4 presents correlations between the change in frequency of parents preparing home-cooked meals and their children’s changes in consumption of different foods during lockdown. Significant, positive correlations were shown for consumption of fruits and vegetables, while there were borderline non-significant correlations for legumes and home-made sweets.

Most participants were interested in healthy nutrition (80.4%) and were getting information in nutrition mainly from either dietitians/nutritionists (60.5%) or the Internet (46.9%). More than half tried to lose weight during lockdown; however, many did not follow a specific diet plan (Table 5).

## 4. Discussion

The COVEAT study was the first study in Greece to explore lifestyle changes in families during the first lockdown and examined associations between changes in parents’ and their children’s behaviour. More parents reported consuming breakfast during lockdown compared to the period before, as well as their children. Parents consumed more snacks during lockdown, mainly sweets, nuts, fruit and vegetables. There was also a significant increase in the frequency of preparing home-cooked meals, which resulted in the reduction of consumption of fast foods. The increase of home-cooked meals also correlated significantly with the children’s increase in the consumption of fruit and vegetables and decreased consumption of fast foods, and there were some indications of a correlation with the increase in home made sweets and legumes. Finally, the interest in healthy nutrition was high during lockdown and some participants tried to lose weight during that period.

Our findings have shown some unfavourable changes in parents’ dietary habits. The increase of snacking was reported by the majority of respondents, and it was a behaviour commonly reported in adult populations during lockdown. A recent scoping review identified several studies which have shown increased snacking in adults during lockdown, which was typically seen at night or after the last meal of the day [7]. Results from the French NutriNet-Santé cohort study in a sample of over 37,000 adult participants showed that 21% reported snacking more than usual during lockdown [18]. Another study in the United Kingdom with 2002 adult participants showed that 56% of the sample increased snacking “a little” or “a lot more” [10].

Apart from frequency, the type of snacking is also important, as it can have a major impact in diet quality and weight changes. In our study, sweets were the most frequently reported snack among parents, similar to cross-sectional studies from Italy, France and Spain, which showed increased frequency (22–50%) of consumption of sweets, such as chocolate, desserts and ice-cream during lockdown [18,19,20].

Boredom, stress, lack of sleep and increased feeling of hunger were the main reasons for parents’ increased snacking in this study. Eating behaviours can be triggered by strong emotions and are initiated through stressful situations, a condition termed as “emotional eating” [21]. Changes in mood and psychological distress are triggers for increased consumption of foods, particularly foods high in calories, fat and/or sugar [21,22]. The World Health Organization has reported that uncertainty about the disease has prompted feelings of anxiety due to the obligation to follow isolation measures and the growing economic wariness [23]. Other studies have also reported increased snacking and consumption of sweets during lockdown [19,24].

Some favourable changes have also been reported in our study, with more parents consuming breakfast and preparing home-made meals, while several increasing consumption of snacks such as nuts, fruit and vegetables during lockdown. The changes in the frequency of breakfast consumption and the effect of home cooked meals in children’s consumption of fast foods, fruit and vegetables further highlights the value of parents as role models into shaping children’s dietary habits. Social environment, and especially parental influence, is known to be a strong determinant of child food behaviour and dietary choices [25]. Increased time at home and interaction between parents and children could serve as an opportunity for parents to influence child behaviour around food. A meta-analysis by Yee et al. [16] provided evidence of the effects of parenting practices, both promotive and preventive, in influencing child behaviour and promoting healthy eating patterns. It should be noted, though, that some of the parents’ favourable changes may be attributed to the higher interest of participants—especially females, who mainly filled in the survey—in healthy eating and to weight loss attempts during confinement.

Consumption of breakfast has been associated with reduced adiposity in children and adolescents [26], and lower body weight [27] and risk of metabolic diseases in adults, such as diabetes mellitus [28] and coronary heart disease [29]. It should be mentioned, however, that meal frequency, total daily energy intake and diet quality all have a major role in weight management and overall health [30], which are factors that were not assessed in the current study.

Cooking and preparation of new recipes might be used as an activity during the lockdown, which in turn could increase the availability of food, especially home-made snacks. The increase in home cooking could be attributable to various reasons including limited opportunities of eating out and/or fast food options, increased available time at home, concerns around safety and a method to deal with increased distress and boredom. About a quarter of adult participants (23%) in Romeo-Arroyo et al.’s study [20] reported that cooking was a skill which they acquired during lockdown and which they intended to keep after confinement. In our study, the correlations between home cooked meals, increased consumption of fruit and vegetables in children and decreased consumption of fast foods in both children and parents, provides some indication of improved diet quality. In a recent, systematic, narrative review by Glanz et al. [31], consumption of home meals was found to be associated with higher diet quality index scores, increased intake of fibre, fruit and vegetables, and decreased intake of sodium, sweetened beverages, unhealthy snacks and fried foods in children and adolescents. The decrease of consuming fast foods during lockdown has been reported in previous studies [24,32,33] and, apart from the increase in home cooked meals, may be related to the closure of restaurants imposed at that time.

Our findings regarding parents’ snacking of nuts, fruit and vegetables are in concordance with other studies, which show increased frequency of consuming vegetables [20,24], fruit [20], fruit juices [24] and nuts [24] in adult populations. Bennett et al. [7] noted that there have been several favourable changes in dietary habits during lockdown, including an increase in the consumption of fresh produce, such as fruit and vegetables.

Increased interest in diet and nutrition and weight loss attempts were reported by the majority of participants, which may be perceived as positive outcomes regarding diet during lockdown. Two recent population studies have also shown similar results in relation to weight loss attempts during lockdown. Brown et al. [34] showed that 79.6% of the study sample tried to lose weight, which was depicted as a “healthy or positive change”. Another UK study with a large sample of 4978 participants showed that nearly half of them (48.5%) tried to lose weight during lockdown [35].

There are strengths and limitations to this study. The COV-EAT study was the first study in Greece to explore dietary changes of parents and potential correlations between parents’ and their children’s dietary changes. Also, the sample was obtained from both urban and rural areas in Greece. Regarding limitations, results may not be generalised to the whole population. Causal relationships could not be determined, due to the cross-sectional design of the study. Data were self-reported and, since some questions required information from the past, the study was subject to recall bias. Selection bias could also influence the data, as the questionnaire was distributed online via networks and social media. It is not known whether part of the study sample was actively seeking nutrition advice (for healthy eating or weight loss) from the dietitians who distributed the survey; however, the high percentage of the sample which reported information from dietitians/nutritionists may indicate selection bias. Also, as the survey required information for only one child of the family, selection bias may have occurred from parents with more than one children. Finally, the questionnaire was predominantly completed by females, which may have influenced the results.

## 5. Conclusions

The study reported both favourable and unfavourable changes in parents’ dietary habits during lockdown, particularly in relation to home cooking (favourable) and snacking (unfavourable/increased). Parents’ dietary changes influenced their child’s eating behaviour as well. Considering the pandemic of COVID-19 is still present and still affects people’s lives, effective strategies and practices are warranted to prevent unhealthy dietary habits, which could further deteriorate health beyond the COVID-19 era.

## Figures and Tables

**Figure 1 children-09-01963-f001:**
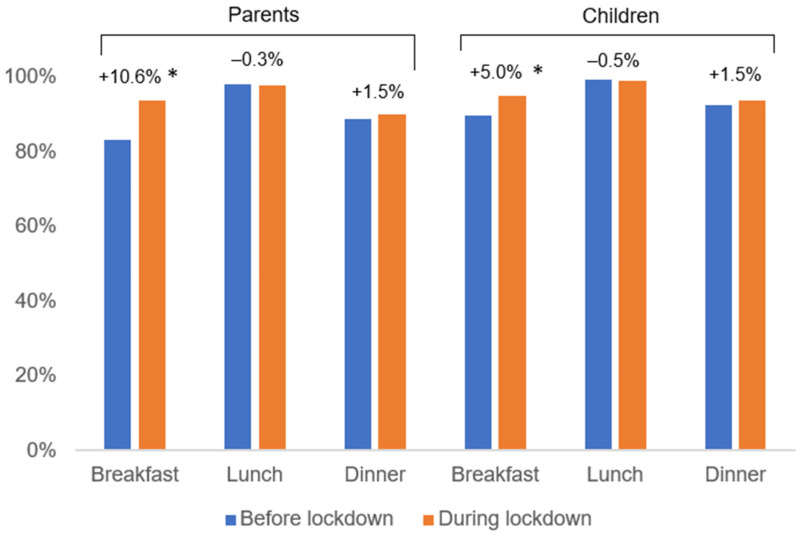
Frequencies of consumption of main meals for parents and their child before and during lockdown and respective changes (*n* 397). * denotes significance (*p* < 0.05).

**Table 1 children-09-01963-t001:** Anthropometric characteristics of the sample.

Characteristics	Mean (SD)
Father’s age (years) (*n* 397)	43.2 (6.4)
Father’s weight (kg) (*n* 387)	88.7 (12.9)
Father’s Body Mass Index (kg/m^2^) (*n* 385)	27.6 (3.8)
Mother’s age (years) (*n* 397)	39.8 (5.3)
Mother’s weight (kg) (*n* 391)	70.1 (14.1)
Mother’s Body Mass Index (kg/m^2^) (*n* 390)	25.6 (4.9)

**Table 2 children-09-01963-t002:** Sociodemographic characteristics of the sample.

Characteristics		N (%)
Area of residence	Urban	331 (83.4)
	Semi-urban	35 (8.8)
	Rural	31 (7.8)
Number of children per household	1	120 (30.2)
	2	221 (55.7)
	3	43 (10.8)
	≥4	13 (3.4)
Father’s education	Primary school	7 (1.8)
	Secondary school	160 (40.3)
	Clerical/Commercial/Professional qualification (IEK)	44 (11.1)
	University Qualification	103 (25.9)
	Postgraduate qualification	83 (20.9)
Mother’s education	Primary school	0 (0)
	Secondary school	125 (31.5)
	Clerical/Commercial/Professional qualification (IEK)	31 (7.8)
	University Qualification	140 (35.3)
	Postgraduate qualification	101 (25.4)
Father’s job before lockdown	Employed	378 (95.2)
	Unemployed	11 (2.8)
	Retired	8 (2.0)
Changes in father’s job during lockdown	No change	181 (45.6)
	Work from home	43 (10.8)
	Increased hours at work	12 (3.0)
	Reduced hours at work	91 (22.9)
	Special purpose leave	32 (8.1)
	Unemployed	11 (2.8)
	Other	27 (6.8)
Mother’s job before lockdown	Employed	311 (78.3)
	Unemployed	84 (21.2)
	Retired	2 (0.5)
Changes in mother’s job during lockdown	No change	166 (41.8)
	Work from home	47 (11.8)
	Increased hours at work	13 (3.3)
	Reduced hours at work	68 (17.1)
	Special purpose leave	55 (13.9)
	Unemployed	16 (4.0)
	Other	32 (8.1)
Family status	Married parents	370 (93.2)
	Single-parent families	27 (6.8)

**Table 3 children-09-01963-t003:** Changes in parents’ food practices and dietary habits (*n* 397).

Practices/Dietary Habits		Before Lockdown	During Lockdown	*p* Value
		N (%)	N (%)	
Consumption of breakfast	Yes	330 (83.1)	372 (93.7)	<0.001
	No	66 (16.6)	23 (5.8)	
	Missing	1 (0.3)	2 (0.5)	
Consumption of lunch	Yes	389 (98.0)	388 (97.7)	0.705
	No	7 (1.8)	7 (1.2)	
	Missing	1 (0.3)	2 (0.5)	
Consumption of dinner	Yes	352 (88.7)	358 (90.2)	0.090
	No	44 (11.1)	37 (9.3	
	Missing	1 (0.3)	2 (0.5)	
Consumption of snacks	No snack	19 (4.8)	7 (1.8)	<0.001
	1	98 (24.7)	59 (14.9)	
	2	248 (62.5)	202 (50.9)	
	3	29 (7.3)	97 (24.4)	
	≥4	3 (0.8)	32 (8.1)	
Preparing meal at home/cooking	Never	4 (1.0)	1 (0.3)	<0.001 *
	1–3 times/month	1 (0.3)	0 (0.0)	
	1–2 times/week	19 (4.8)	4 (1.0)	
	3–4 times/week	73 (18.4)	30 (7.6)	
	5–6 times/week	98 (24.7)	71 (17.9)	
	Every day	202 (50.9)	291 (73.3)	
Person who cooks meals at home ^1^	Mother	337 (84.9)	369 (92.9)	<0.001
	Father	41 (10.3)	61 (15.4)	0.001
	Grandparents	116 (29.2)	52 (13.1)	<0.001
	Other	9 (2.3)	3 (0.8)	
Ordering fast food	Never	23 (5.8)	202 (50.9)	<0.001 *
	1–3 times/month	203 (51.1)	126 (31.7)	
	1–2 times/week	141 (35.5)	50 (12.6)	
	3–4 times/week	26 (6.5)	13 (3.3)	
	5–6 times/week	2 (0.5)	4 (1.0)	
	Every day	2 (0.5)	2 (0.5)	
Provision of dietary supplements to children ^1^	No supplements	301 (75.8)	295 (74.3)	0.239
	Multivitamin	28 (7.1)	30 (7.6)	0.564
	D	33 (8.3)	39 (9.8)	0.033
	C	10 (2.5)	14 (3.5)	0.102
	Omega-3	33 (8.3)	33 (8.3)	1.000
	Other	14 (3.5)	16 (4.0)	

^1^ More than one option. * Paired *t*-tests for continuous variables, as mentioned in text (transformed from nominal variables).

**Table 4 children-09-01963-t004:** Correlations of changes in frequency of preparing home-cooked meals and child’s consumption of food groups (*n* 397).

	Food Groups (Child)	Spearman’s Rho	*p* Value
Home-cooked meals	Fast foods	−0.192	<0.001
	Vegetables	0.106	0.036
	Fruit	0.137	0.006
	Red meat	−0.002	0.963
	Poultry	−0.076	0.132
	Fish	0.002	0.696
	Pasta	0.038	0.446
	Legumes	0.094	0.062
	Home-made sweets	0.098	0.052
	Ready-made sweets	−0.036	0.470
	Salty snacks	0.026	0.601
	Sodas	−0.065	0.196

**Table 5 children-09-01963-t005:** Interest in healthy eating and weight loss practices.

Characteristics		N (%)
Are you interested in healthy eating? (*n* 397)	Yes	319 (80.4)
	No	78 (19.6)
Main source of information (*n* 324)	Internet	186 (46.9)
	Newspaper/magazines	35 (8.8)
	Television	24 (6.0)
	Dietitian/Nutritionist	240 (60.5)
	Other	14 (3.5)
Are you trying to lose weight? (*n* 397)	Yes	232 (58.4)
	No	165 (41.6)
Do you follow a specific plan? (*n* 397)	Yes	150 (37.8)
	No	247 (62.2)
When did you start you weight loss plan? (*n* 183)	Before lockdown	89 (48.6)
	After lockdown	94 (51.4)

## Data Availability

The data presented in this study are available on request from the corresponding author. The data are not publicly available due to ethical restrictions.

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
