# Peer review of "Parental Lifestyle Changes and Correlations with Children’s Dietary Changes during the First COVID-19 Lockdown in Greece: The COV-EAT Study"

_children, 2022, doi:10.3390/children9121963_

Round 1

Reviewer 1 Report

Thank you for the opportunity to review "Parental lifestyle changes and correlations with children's dietary changes during the first COVID-19 lockdown in Greece: The COV-EAT study.  This is a well-written paper that presents their story.

Some comments:

Overall

The paper does not address or explain why dietitian/nutritionist networks were used to distribute the questionnaire and what impact this might have on the results.  It does not discuss whether the parents were actively seeking treatment from the dietitian/nutritionist, at the time of the survey or before it.  Using these networks would directly impact the very high levels of people reporting getting their dietary advice from dietitians/nutritionists. These levels were significantly higher than what is usually reported by the general population.  It is also likely impacting the number of people actively trying to lose weight during the pandemic.  It certainly appears to create bias, with a potentially greater proportion of people reporting healthier changes than you might have expected with a general population. Although the bias is generally mentioned in the limitations, further exploration/ explanation of this bias should be included.

Methods

Under the heading Instruments and variables

This section requires greater detail. It is not clear if the answers were free text or multiple choice and if they were multiple choice, what were the choices. It was also unclear if the researchers or the participants decided what change occurred and how "stable", "increased", or "decreased" were defined. Although greater detail is explained in (Androutsos et al., 2021), I believe this paper would benefit from greater detail in this section.

It is unclear how the parents decided on what child they would base their answers on, and there was no discussion on how this may or may not bias results.

Results

The tables are formatted in a way that is very difficult to read some lines or spaces between variables would aid in their readability.

Please consider adding significant differences to Figure 1.

Discussion

In lines 201-202 do you mean to say "…a major impact on diet quality and weight changes. "rather than effect?

Limitations needs greater exploration see above

Reviewer 2 Report

This article is one of the articles which had to intention to show changes in the eating behaviour of parents during the first lockdown implemented due to COVID-19 and to explore possible associations with corresponding changes in the eating behaviour of their children. It includet 397 parents with children aged 2-18 years in urban and rural parts of Greece. It showed interesting positive results regarding eating breakfast and home-cooking meals with decreased consumption of fast foods for both parents and children and with increased consumption of fruit and vegetables for children. Study concluded, both favourable (home-cooking) and unfavourable (increased snacking) lifestyle changes during the first COVID-19 lockdown for parents.

English language

I would like to say that I am not a native English speaker so maybe I am not enough competent to comment on english, so my question would be did you have certificate language check up of your article?

Review

 Study is well designed, everything is clear,  I do not have any doubts or additional questions. I think that this study can show us on what we need to work in future so that children become healthy adults

Author Response

Dear reviewer, 

Thank you for your comments. We did not use a language check for this manuscript. The manuscript has been prepared by a team who is proficient in English. 

With regards, 

Dr Georgios Saltaouras, on behalf of Prof Odysseas Androutsos